# Extrinsic nature of the broad photoluminescence in lead iodide-based Ruddlesden–Popper perovskites

Simon Kahmann [1], Eelco K. Tekelenburg [1], Herman Duim[1], Machteld E. Kamminga [1,2] & Maria A. Loi [1✉]

Two-dimensional metal halide perovskites of Ruddlesden–Popper type have recently moved into the centre of attention of perovskite research due to their potential for light generation and for stabilisation of their 3D counterparts. It has become widespread in the field to attribute broad luminescence with a large Stokes shift to self-trapped excitons, forming due to strong carrier–phonon interactions in these compounds. Contrarily, by investigating the behaviour of two types of lead-iodide based single crystals, we here highlight the extrinsic origin of their broad band emission. As shown by below-gap excitation, in-gap states in the crystal bulk are responsible for the broad emission. With this insight, we further the understanding of the emission properties of low-dimensional perovskites and question the generality of the attribution of broad band emission in metal halide perovskite and related compounds to self-trapped excitons.

[1] Zernike Institute for Advanced Materials, University of Groningen, Nijenborgh 4, 9747 AG Groningen, The Netherlands. [2] Present address: Department of Chemistry, Inorganic Chemistry Laboratory, University of Oxford, South Parks Road, Oxford OX1 3QR, UK. ✉email: m.a.loi@rug.nl

Low-dimensional organic–inorganic metal halide perovskites are a class of materials with interesting opto-electronic properties relevant for a variety of applications. Although the 2D compounds have been the subject of research since the 1980s[1,2], the recent advent of their 3D counterparts for use in photovoltaics led to a surge of research also into low-dimensional perovskite-like structures. In their 2D variants, inorganic slabs of metal halide octahedra are sandwiched between long organic spacer molecules. This creates both a quantum and a dielectric confinement of charge carriers into the inorganic layers resulting in an exciton binding energy of around 300 meV (depending on the composition) and concomitant rich exciton physics.

The luminescence of these compounds is thus commonly dominated by a narrow emission (NE, full width at half maximum, FWHM ~10 meV) attributed to the recombination of free excitons (FE)[2]. Such NE is beneficial, e.g., for highly colour-pure red–green–blue (RGB) applications, as results for light emitting diodes have demonstrated[3]. Recently, though, large interest has been directed towards compounds that exhibit (sometimes additionally) a broad and strongly Stokes-shifted emission band (BE, FWHM larger than 100 meV), which could offer the opportunity for direct white-light generation[4–9]. It has become common in the community to attribute the latter to the emission from so-called self-trapped excitons (STEs) formed due to the strong carrier–phonon interaction in these materials[10].

In particular, a variety of chloride and bromide-based 2D perovskites have been shown to either give a combination of the NE and BE or to only show the latter[5,9,11–13]. STEs have previously been invoked for the broad emission bands of a variety of materials[14–18], but it is probably their observation in lead halides[19,20] and layered alkylammonium cadmium chlorides[21] that made researchers make the connection towards the current materials.

There is indeed significant theoretical work addressing STEs in this material class and often the distortion and torsion of the inorganic cages are proposed to be the major source of STE formation[22–24]. Nonetheless, some controversy remains. Lead-iodide-based compounds ($A_2PbI_4$), for example, predominantly exhibit spectra containing only the NE of FEs[2,25,26]. Only few groups reported a weak broad emission for PbI-based thin films, commonly at low temperature[27–29]. The actual role of STEs in these materials thus remains unclear and the question stands if they are responsible at all.

To address this controversy, we closely examine the behaviour of two different types of lead iodide-based 2D perovskites (phenylethylammonium (PEA) lead iodide, $PEA_2PbI_4$ and 3-fluorophenylethylammonium iodide, $FPEA_2PbI_4$), which are both able to show a broad emission band at room temperature. In contrast to most reports, we focus on single crystals in order to exclude grain boundaries and interfaces as the origin of the BE. Through a combination of spectroscopic techniques, we show that intrinsic self-trapping is not responsible for the broad emission in these materials. Instead, we reveal that extrinsic factors determine the presence or absence of the broad emission band and we relate them to in-gap states caused by defects. Temperature-dependent analysis and low-energy excitation are used to highlight that several emitting states are present within these materials' band gap. We hence show that exciton self-trapping and its importance in halide perovskites cannot be postulated a priori, but its relevance has to be demonstrated experimentally case by case. In this light, we aim to spark a discussion on and a possible re-evaluation of previous results on related compounds and present easy strategies to do so.

## Results

**Room temperature luminescence.** STEs can be considered similarly to small polarons, but involving both carrier types. Put

simply, due to a strong interaction with phonons of the surrounding material, excitons localise and become trapped in a potential well created by their own presence[30]. The actual formation pathway can vary, but it is often assumed that one carrier type localises to form a small polaron, which subsequently attracts its countercharge[31,32]. This behaviour is thus intrinsic to perfect, defect-free materials. Typically, these states are discussed and illustrated by using a configurational coordinate scheme as depicted in Fig. 1a. For an adiabatic excitation from the relaxed ground state (GS), either FE or free carriers are formed (blue arrow). In 2D perovskites, the latter quickly form excitons. The above discussed deformation and stabilisation of the excitons' environment manifest themselves in a large change of the configurational coordinate ($Q$, typically discussed in terms of the Huang–Rhys factor, $S$). The energy difference between the FE and STE state is the relaxation energy $E_r$, which, together with $S$, leads to the large Stokes shift between FE (green) and STE emission (red). As indicated, self-trapping generally requires excitons to overcome an energy barrier $E_t$ (trapping energy) whilst the detrapping of STEs demands for overcoming $E_{dt}$. Free excitons and STEs can thus coexist in the same material, leading to the observation of both NE and BE bands[10].

Notably, the above discussion treats the case of so-called intrinsic self-trapping. Materials that exhibit an exciton–phonon

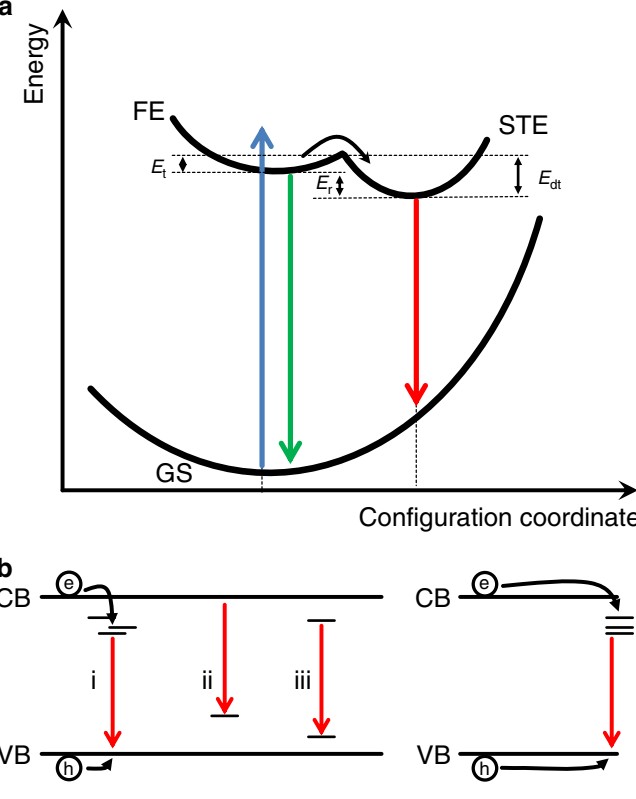

**Fig. 1 Photophysical processes leading to low-energy emission.** Free (FE) and self-trapped excitons (STE) shown in a configuration coordinate model (**a**). Excitation from the ground state (GS) generates FEs or free carriers (not shown), which can self-trap when overcoming a barrier $E_t$. The concomitant lattice deformation reduces the minimum of the STE by the relaxation energy $E_r$ with respect to FEs. STEs can also become FEs again, when a detrapping barrier $E_{dt}$ is overcome. Defect states within the bulk (left) or at the surface (right) of a material can also evoke low-energy luminescence (**b**). This could be due to the recombination of trapped with free carriers (i, ii) or through the recombination of two trapped carriers (iii).

coupling too weak to create STEs themselves might nonetheless give rise to so-called extrinsic STEs. Typically, this is the case when small amounts of isoelectronic impurities are added and local potential wells are created in a host material. These potential wells lead to an initial carrier localisation, upon which the exciton–phonon interaction is strong enough for the exciton to self-trap[30,33–35].

Luminescence at energy lower than the fundamental band gap can also be due to in-gap states acting as traps. As illustrated in Fig. 1b, the recombination of a carrier trapped in such a state with a free carrier (i, ii) or the recombination of a so-called donor–acceptor pair (iii) can, if radiative, lead to the emission of a photon of below-band-gap energy[30]. It is important to note that broad bands might arise due to an energetically broad distribution of trap states (illustrated in i), but also due to strong coupling to the lattice (large S). Sketches as shown in (b) commonly neglect the latter aspect since the interaction of defects with classical semiconductors, such as silicon or GaAs is often negligible (small S). Nonetheless, especially deep trap states can strongly couple to phonons in their host[36]. Additional to states in the bulk, defects will always occur at the surface of a material and can lead to radiative recombination, as shown on the right in Fig. 1b.

In our study, we examined two different sets of Ruddlesden–Popper single crystals based on lead iodide ($A_2PbI_4$) with either the commonly employed PEA or its fluorinated derivative 3-fluorophenylethylammonium (FPEA) as organic spacer molecule in the A-position. For convenience, we shall refer to the two materials solely through their organic cation. A sketch for the crystal structure is illustrated in Fig. 2a. Flakes were cleaved prior to each measurement giving rise to smooth and fresh surfaces, as seen in the wide-field transmission microscopy image (b) or wide-field PL image (c) of Fig. 2 (also consider Supplementary Fig. 1). The A-site molecules differ only by the fluorine/hydrogen atom in the meta position of the phenyl ring, as shown in the inset of Fig. 2d. Fluorination of spacer molecules was recently used to increase the hydrophobicity of 2D perovskites in order to improve the material stability towards moisture — a pressing topic in perovskite-based device research[37,38]. Here we focus on the fact that in both cases, fabricated single crystals exhibit a generally strong narrow PL band and an additional broad emission at lower energy (Fig. 2d), while a separate study on the impact of fluorination on the general photophysics is in preparation. The two emission bands scale approximately linearly over a wide range of excitation intensity (Supplementary Fig. 2) indicating first order recombination (Supplementary Note 1). Notably, whilst data on FPEA has so far been scarce[39], PEA typically only exhibits the NE at room temperature[26,40].

The spectra in Fig. 2d were obtained by exciting the sample in free space and focusing the laser to a spot diameter of around 100 μm on the crystals. The obtained spectra exhibited a large point-to-point variation in terms of the NE and BE intensities, which furthermore strongly depended on the specific crystal flake under consideration. Some appeared bright green (green flake) whereas others exhibited a distinctively red emission (red flake), as illustrated by the wide-field PL images reported in Supplementary Fig. 3. This colour variation could be directly linked to a variation of the intensity of the narrow and broad emission bands. Bright green flakes hardly showed any contribution of the BE and red flakes not only offered a much more pronounced BE, but also exhibited a drastic reduction in NE.

Using confocal laser scanning microscopy (CLSM), the crystals were subsequently investigated in more detail (lateral resolution of up to 200 nm). We mapped the luminescence and filtered it for two distinct spectral regions: from 2.28 to 2.56 eV (green channel) and below 1.9 eV (red channel), i.e., perfectly coinciding with the

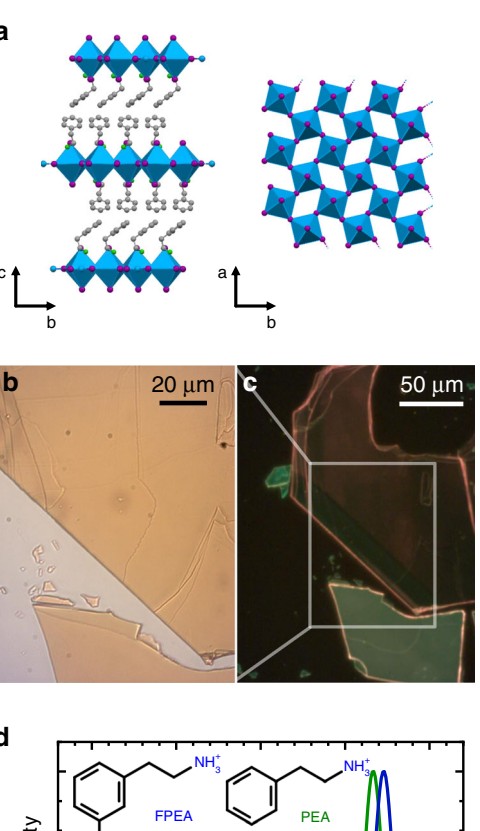

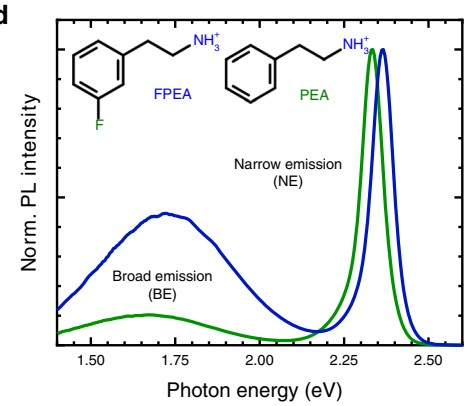

**Fig. 2 Single crystal characterisation.** Crystal structure of the employed RP-type perovskite along high symmetry directions (**a**) (PEA is shown). Slabs of inorganic $PbI_6$ octahedra are sandwiched between double layers of (F)PEA ions, as seen on the left. The structure on the right only contains the inorganic octahedra, illustrating their twisted arrangement. The wide-field transmission microscopy image (**b**) and wide-field PL image (**c**) highlight the smooth surface of exfoliated single crystals. Waveguiding brightens the crystal edges in **c**. For both compounds, the PL spectra (**d**) show a combination of a pronounced narrow feature around 2.35 eV with a broad emission at 1.7 eV. The inset depicts the structure of the two molecular ions.

NE and BE. A particularly illustrative example for FPEA is given in Fig. 3 (for clarity, we focus the discussion in the main text on data obtained from FPEA and show PEA data, which are analogous, in Supplementary Fig. 4). The image in (a) shows the confocal PL map detected through the green channel and (b) displays the same area measured through the red channel. A striking contrast emerges for example when considering the areas denoted A and B. Whilst the former exhibits bright green emission, the red channel intensity in this spot is weak. On the other hand, B displays high intensity red emission over the entire flake where the green channel's signal is only faintly visible. Representative spectra for points A and B are given in (c) illustrating the trade-off between NE and BE for the two areas. In

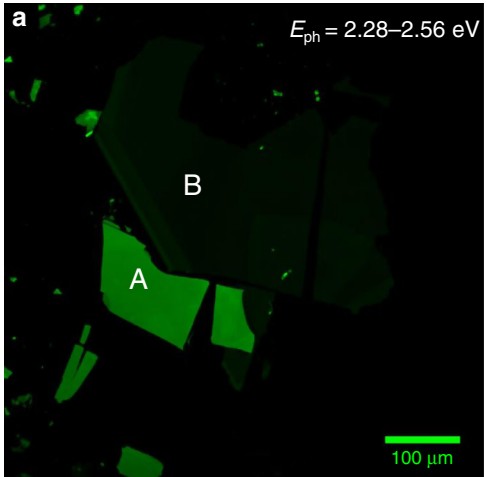

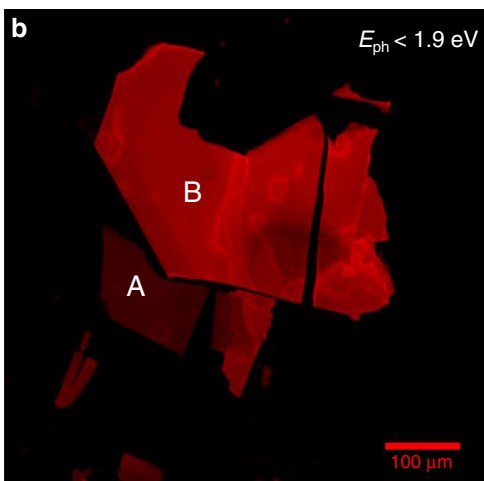

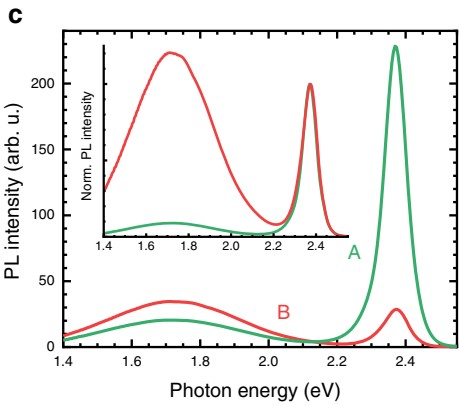

**Fig. 3 Confocal PL maps of FPEA single crystals.** The green (photon energy $E_{ph}$ of 2.28–2.56 eV) and red (smaller than 1.9 eV) channel shows complementary intensity maps with bright green areas in **a** appearing especially dim in **b** and vice versa. Representative spectra for such points are given in **c**, indicating the pronounced FE in A and much stronger BE in B. The inset highlights the intensity differences through normalisation to the NE peak.

general, both types of crystals exhibit a relatively large flake-to-flake variation of the NE:BE ratio and in some cases also large differences within single flakes. Importantly, none of the investigated samples exhibited the red-shifted emission solely on the crystal edges (Supplementary Figs. 3–5), as proposed elsewhere[41]. This also serves to exclude thickness effects to lie at the origin of the different colour impression.

We used a combination of experiments discussed in Supplementary Note 2 (Supplementary Figs. 5–9) to furthermore ascertain that the BE is not due to surface processes, if not the edges, and indeed find that the origin must lie within the bulk of the crystal flakes.

Prolonged illumination on the scale of minutes significantly reduces the BE (Supplementary Figs. 10, S11), which excludes it to be due to light-induced defects as previously speculated[42]. Simultaneously, the NE shows some minor variation, but by far not as pronounced as the BE. Importantly, the NE and BE behaviour upon illumination do not correlate. Underlining this point, we also measured the time-resolved PL of green and red flakes (Supplementary Note 4). The data in Supplementary Fig. 12 reveals that the NE exhibits a significantly longer lifetime in green areas than in areas of pronounced BE. In accordance with the steady-state data, prolonged illumination slightly increases the NE lifetime, but not towards values found for green flakes.

These measurements, together with the PL heterogeneity shown in Fig. 3, strongly suggest an extrinsic origin of the BE and we would like to highlight that we showed in a parallel study on thin films how the BE can be suppressed or increased through simple variation of the precursor stoichiometry (Supplementary Fig. 20)[43]. The photo-activated suppression of the BE additionally hints to the role of mobile species in the material, which we shall further comment on below.

**Temperature-dependent behaviour.** Having established that both materials exhibit large variations in luminescence due to extrinsic effects in the bulk, we aim to further elucidate the behaviour of the two emitting states through temperature-dependent photoluminescence. Figure 4 shows the normalised PL spectra in a two-dimensional false-colour plot for a red (a) and a green (b) FPEA flake with spectra at significant temperatures extracted in (c) and (d) (for the complete overview see Supplementary Figs. 13 and 17 for data on PEA). Strikingly, the BE becomes very strong in all cases upon temperature reduction. Cooling down leads to a narrowing of all involved signals and below ~80 K the NE clearly reveals its substructure. At the same time, the BE peak intensity around 1.75 eV decreases below 80 K and a third distinct emission band (I) forms in an intermediate region around 2.15 eV (further discussion along Supplementary Figs. 13–17, Supplementary Note 4). The different trends in PL intensity along with the presence of I and a clear substructure also of the BE at higher temperature shows that the obtained luminescence spectra cannot simply be explained by a two-state model and the samples exhibit a complex interplay of emissive states and recombination pathways. The revealed distinct differences for the green and red flakes also further underline the extrinsic nature of the BE. As a side note, we would like to mention that both types of materials can exhibit PL peaks around 2.5 eV, on the high energy side of the NE (Fig. 4c, d and Supplementary Fig. 13). These peaks are different from the previously reported hot PL peaks[27] and we tentatively attribute them to small amounts of $PbI_2$[44,45], which can form through extended illumination[26].

**Below-gap excitation.** Whilst all observations above point to defects as responsible for the BE, one might still argue extrinsic STEs to be the main cause of the BE and the other observations to be unrelated side-effects. In order to discriminate between the two effects, we considered the following: STEs are transient species forming due to the deformation of the crystal lattice — they cannot be excited directly, but are the result of the interaction of phonons with FEs (consider Fig. 1a, discussion above). Above-gap excitation thus often leads to luminescence

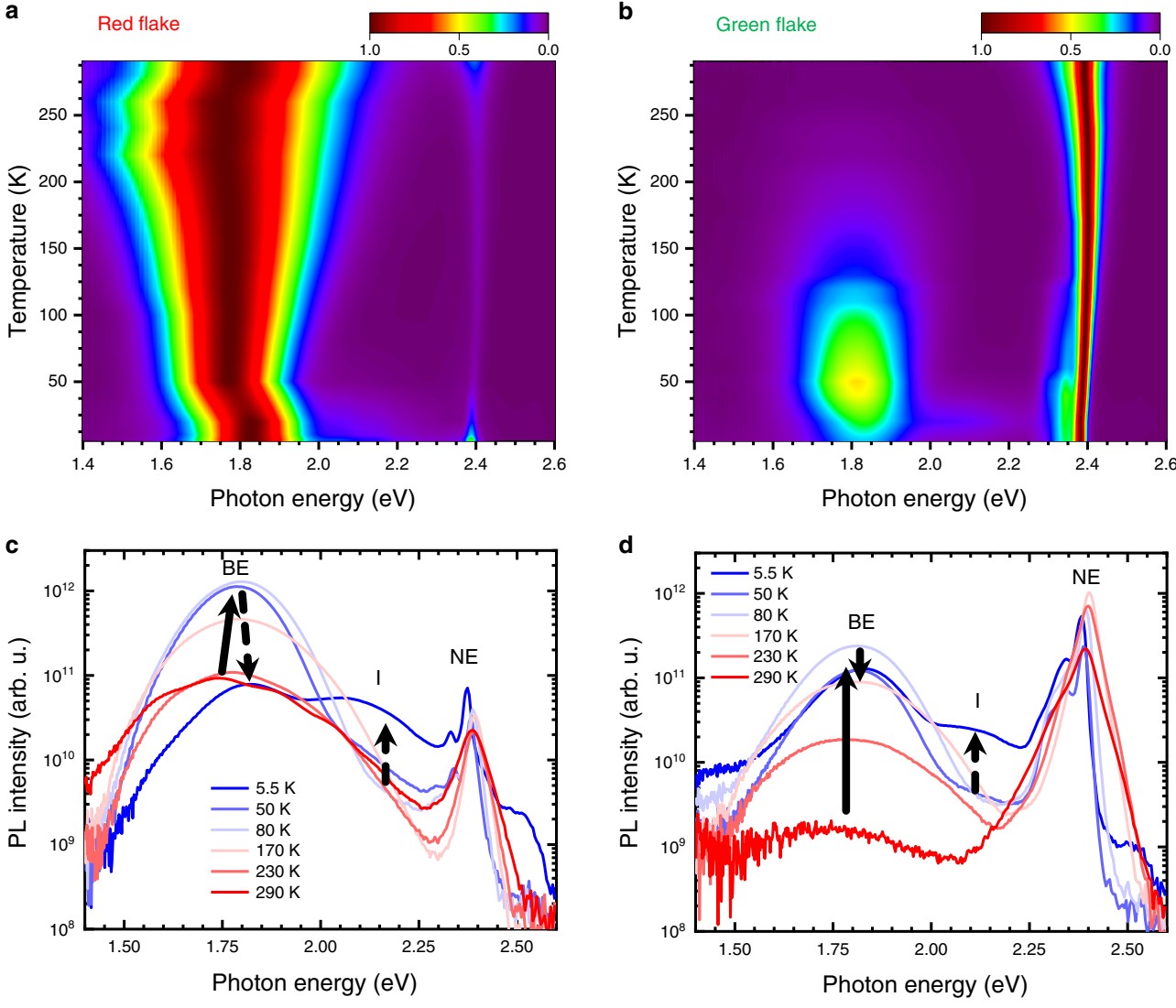

**Fig. 4 Temperature-dependent PL spectra.** Normalised false-colour plots of the PL from FPEA crystals with (**a**) and without (**b**) significant BE at room temperature. The extracted spectra are given in **c** and **d**. Low-temperature spectra reveal a finite emission from an intermediate state *I* around 2.15 eV especially for the red flake (**a, c**).

from both states (Fig. 5a, left). In-gap states, on the other hand, can be excited directly from the GS (Fig. 5a, centre). We therefore used photons of below-band-gap energy to probe for possible photoluminescence. When carrying out such an experiment, it is important to note that perovskites have a large non-linear absorption cross-section leading to strong two-photon absorption[40]. However, this process predominantly occurs for high photon fluence and scales quadratically with the incident photon density (Fig. 5a, right).

For a number of different excitation energies below the excitonic resonance, we observe identical luminescence spectra from the BE shown in Fig. 5b. We additionally verified the absence of two-photon processes by measuring the power dependent PL. Figure 5c shows how below-gap excitation at 2.07 eV with a picosecond source[46] exhibits a linear dependence of the BE (intensity at 1.75 eV, 710 nm), while high intensity femtosecond pulses at 1.55 eV lead to a quadratic dependence of both the narrow and broad emission intensity (for additional data see Supplementary Figs. 18, 19 and Supplementary Note 5). The data in Fig. 5b, c thus serve to show that the BE is not due to STEs, but due to in-gap states.

## Discussion

One surprising aspect of our samples is the large flake-to-flake variation of the BE intensity as well as the common observation of only the NE in previous reports on PEA-based samples (at room temperature). As we demonstrated in our recent work on thin films, the precursor ratio has a strong impact on the luminescence properties of these materials (Supplementary Fig. 20)[43]. Cast from solutions with an excess of PEAI, thin films will show the BE band. This observation is also in line with a separate study on $PEA_2PbI_4$, in which crystals were synthesised under varying amounts of iodide precursor that strongly affects the magnitude of the BE[47]. We thus assume that local concentration variations or precursor depletion during the crystal growth is responsible for the differences in our samples.

The presence of sub-peaks of the broad emission, as well as what we termed intermediate state *I* underlines that several states can form in the fundamental band gap of the investigated compounds. We see a clear resemblance of this defect-related luminescence to the cases of, for example, GaN, ZnO, or even $PbI_2$[48–53]. These materials often exhibit a NE peak with one or more additional broad and strongly Stokes-shifted bands at lower

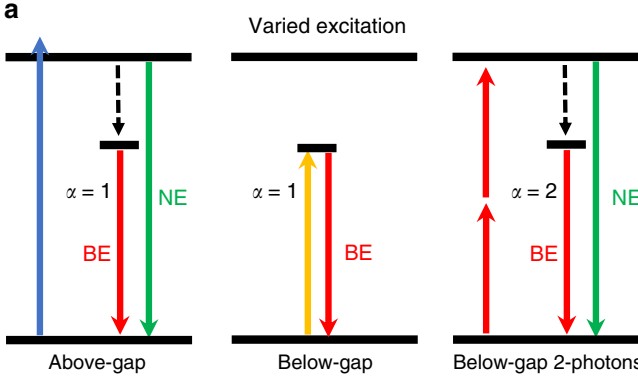

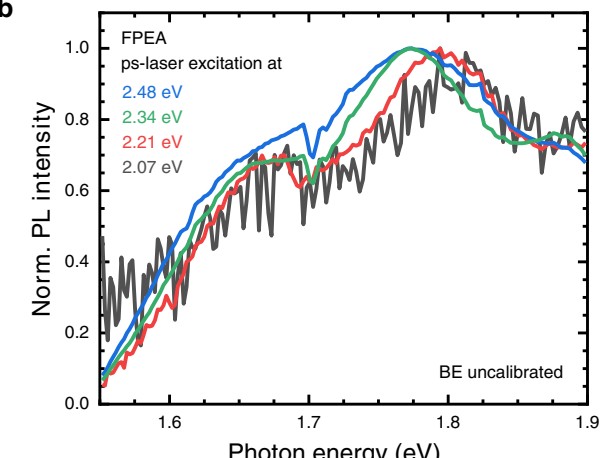

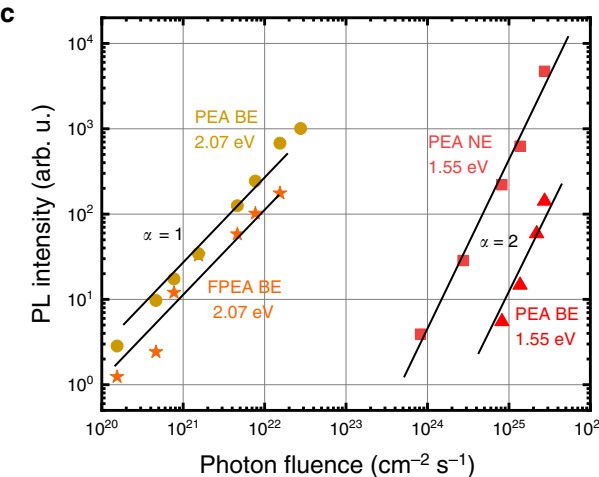

**Fig. 5 Below-gap excitation.** Two-photon excitation scales quadratically with the incident intensity whilst direct excitation scales linearly (**a**). The broad emission can be excited by using photons whose energy lies below fundamental band gap (**b**) and the detected PL intensity scales linearly with the incident photon fluence at 2.07 eV (600 nm) for both types of flakes (**c**). This clearly contrasts with the behaviour upon 1.55 eV (800 nm) excitation, for which direct excitation is impossible.

energy. Strikingly, these materials exhibit a relatively large carrier–phonon interaction but exciton self-trapping does not play a role[34].

In this light, we find it important to stress that although metal halide perovskites and their related compounds generally exhibit impressive opto-electronic performance metrics, it should be no surprise that some of their characteristics are governed by defects.

The impact of defects should become particularly evident when considering low-temperature phenomena. In this context one should furthermore note that different molecular ions and halides exhibit different reactivity. If casting protocols have not been properly optimised, differences in luminescence properties could thus easily be due to slight differences in stoichiometry of fabricated samples[43]. This could also explain the contrast between chloride and bromide-based materials, which are often reported with pronounced BE at room temperature, whilst few cases of BE for iodide-based compounds have been discussed so far.

Besides a possible involvement of impurities, the dynamic behaviour of the BE upon extended illumination points to another aspect of halide perovskites. Especially for the 3D counterparts of these materials, the (light-mediated) diffusion of ions, particularly methylammonium and halide ions, has been associated with changes in luminescence spectra and dynamics[26,54,55]. Given the absence of the former, we consider it probable, that iodide-related (more generally, halide-related) defects, such as vacancies or interstitials play a decisive role also in the 2D compounds. The latter, in particular, were proposed to form deep states[56] and were previously linked to the broad emission[47].

Experimental and theoretical approaches to design broadly emitting perovskite compounds with high quantum yield thus have to be aware of this distinction. Distortions of the inorganic cages, for example, might very well be a way for efficient STE emission in some compounds, but reducing the discussion in metal halide perovskites to such effects will fail to catch the full physics of the carrier recombination.

In summary, we investigate two different sets of single crystals composed of lead iodide-based 2D perovskites. Both the commonly employed PEA and its fluorinated variant FPEA exhibit a NE peak at high energy and a red-shifted broad emission band. Also, both sets of crystals exhibit a large flake-to-flake variation of the luminescence spectra. We therefore use a combination of optical spectroscopy techniques to show that the increasingly common attribution of any broad emission to be due to STEs is incorrect. In contrast, low temperature and below-band-gap excitation experiments show that this band is affected by external factors—an observation in line with our work on thin films. In-gap trap states in the bulk of the crystals are responsible for the broad emission and give rise to complex dependencies of the luminescence intensity of our samples. This defect-governed luminescence strongly resembles the green, yellow, and red bands known from ZnO, GaN, or $PbI_2$.

Our findings have important consequences for the design of broadly emitting perovskite-based materials, where the common assumption is that exciton self-trapping is solely responsible for bright and Stokes-shifted emission bands.

## Methods

**Synthesis**. Single crystals of $[3-FC_6H_4CH_2CH_2NH_3]_2PbI_4$ were grown at room temperature using a modified version[57] of the layered-solution approach, previously reported by Mitzi[58]. In this method, $3-FC_6H_4CH_2CH_2NH_2$ and $PbI_2$ are dissolved in separate solutions of different densities. Due to the density difference between the solutions, a sharp interface is formed when the two components are brought together. As slow diffusion takes place at the interface, single crystals start to grow.

The single crystals were synthesised by closely following the procedure described in our previous work[57]. Here, 74 mg (0.16 mmol) $PbI_2$ (Sigma-Aldrich; 99%) were dissolved in 3.0 mL of concentrated (57 wt%) aqueous hydriodic acid (Sigma-Aldrich; 99.95%). 3.0 mL of absolute methanol (Lab-Scan; anhydrous; 99.8%) were placed on top of the light-yellow $PbI_2$/HI mixture, without mixing both solutions. Due to the vast difference in densities (0.791 g mL$^{-1}$ for methanol and 1.701 g mL$^{-1}$ for the concentrated aqueous hydriodic acid), a sharp interface between the two layers forms. $3-FC_6H_4CH_2CH_2NH_3$ (Sigma-Aldrich; 99%) was added in great excess by gently adding 15 droplets on top of the methanol layer. Small crystals formed at the interface and subsequently gathered at the bottom of the tube. After 5 days, the crystals were extracted and washed with diethyl ether

(Avantor). Following drying under ambient conditions, the crystals were stored in a nitrogen glove box. The crystals are bright orange in colour and shaped like platelets, with the biggest crystals being around 2 mm across.

Single crystals of PEA iodide were grown in an almost identical synthesis method. Here, 15 droplets of 2-phenethylamine (Sigma-Aldrich; 99%) were used to replace the fluorinated organic molecule described above.

**Crystal cleaving.** The crystal flakes were exfoliated using a "sticky tape method." Here, a crystal is placed on a piece of sticky tape and the tape is repeatedly folded and unfolded on top of the crystal. During this process layers of the 2D perovskite are peeled off and cover the entire tape. By pressing a $1 \times 1$ cm quartz substrate firmly onto the tape, small flakes are transferred onto the substrate.

**Photoluminescence spectroscopy.** The crystals were mounted into a cryostat (Oxford Instruments Hires or Optistat CF) and excited at 3.1 eV (400 nm) using the second harmonic of a mode-locked Ti:sapphire laser (Mira 900, coherent) at a repetition rate of 76 MHz. Steady-state spectra were recorded with a Hamamatsu EM-CCD camera that was spectrally calibrated. The excitation beam was spatially limited by an iris and focused with a lens of 150 mm focal length. The fluence was adjusted using grey filters and spectra were taken in reflection geometry. Time-resolved traces were taken with a Hamamatsu streak camera working either in synchroscan or single sweep mode. An optical pulse selector was used to vary the repetition rate of the exciting pulses where necessary. At room temperature, measurements occurred under nitrogen unless stated otherwise, and a helium atmosphere was used for the experiments at low temperature.

Low fluence excitation below the band gap was performed using a supercontinuum laser (NKT Photonics SuperK Extreme, 76 MHz repetition rate, 1–2 ps pulse duration) as excitation source and the photoluminescence[46] was detected through a spectrograph (Shamrock SR303i, Andor) with 150 l mm$^{-1}$ grating (800 nm blaze) equipped with an EM-CCD camera (Luca R, Andor).

**Microscopy.** A microscope set-up, based on a Nikon Eclipse Ti, was used to map the photoluminescence of the crystal flakes. Bright field micrographs were taken with a CMOS camera (Basler PowerPack) using a white-light source in transmission geometry. CLSM was performed in reflection geometry using a single-line continuous wave laser (488 nm, Melles Griot) and a set of photomultiplier tubes to detect the signal in two spectral ranges ($515 \pm 30$ and >650 nm). For each measurement, a z-stack is made by projecting slices of the micrographs in z direction on top of each other to extend the depth of focus. By coupling the femtosecond laser through a multi-mode optical fibre into the confocal microscope and collecting the luminescence through a second fibre, time- and spatially-resolved photoluminescence spectroscopy could be performed with a lateral resolution of up to 200 nm.

**Below-band-gap excitation.** In order to probe for photoluminescence upon excitation with photons below the band gap, the crystals were illuminated at different wavelengths emitted by the supercontinuum laser. Appropriate band pass filters were used in the excitation beam path to assure colour purity. Significant laser scattering required the use of filters also in the collection beam path. There were long-pass filters that allowed for tracking the low-energy region of the BE, but cut off the high energy part. Fluence-dependent measurements were carried out with a set of two 610 nm and one 645 nm long-pass filters.

Excitation at 1.55 eV (800 nm) was carried out with the Ti-sapphire-based set-up discussed above. The 800 nm fundamental emission was used to excite the crystals and the strong laser scattering was suppressed by a high-quality 775-nm short-pass filter (optical density 4) to allow for tracking the BE.

## Data availability
The data that support the findings of this study are available from the corresponding author upon reasonable request.

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

## Acknowledgements

S.K. acknowledges the Deutsche Forschungsgemeinschaft (DFG) for a postdoctoral research fellowship (grant number 408012143). This work was financed through the Materials for Sustainability (Mat4Sus) programme (739.017.005) of the Netherlands Organisation for Scientific Research (NWO) and is a publication of the FOM-focus Group "Next Generation Organic Photovoltaics," participating in the Dutch Institute for Fundamental Energy Research (DIFFER). M.E.K. was supported by The Netherlands Organisation or Scientific Research (NWO) (Graduate Programme 2013, No. 022.005.006). Arjen Kamp and Teo Zaharia are kindly thanked for technical support.

## Author contributions

M.E.K. synthesised the materials. E.K.T. and H.D. carried out the confocal laser scanning microscopy. E.K.T. and S.K. performed all other experiments. The work was supervised by M.A.L. S.K. conceived the study and wrote the first draft. All authors contributed to the final version of the manuscript.

## Competing interests

The authors declare no competing interests.
