## [Peer Review File · Nature Communications]

Reviewers' comments:

Reviewer #1 (Remarks to the Author):

Loi & co-workers report an interesting and carefully-executed study of broad emission (BE) in two A₂PbI₄ perovskites: PEA₂PbI₄ and a mono-fluorinated variant thereof ("FPEA"). Through a wide array of photoluminescence characterization techniques, they conclude that BE in FPEA originates from a distribution of static defect states – rather than a self-trapped exciton (STE).

I agree with the authors' assignment of broad emission in their FPEA samples to defect states and commend them on their careful spectroscopic characterization. However, the findings reported in this manuscript are not nearly generalizable enough to call into question the STE assignment of temperature-dependent BE in other 2D lead halide perovskites. (I want the authors to know that I write this as someone intimately familiar with the spectroscopic characterization of these materials but who has not previously published on STEs; I have no stake in the assignment of the broad spectral feature either way). Once the claims of the paper are scaled back, it could be published in a more specialized journal. However, it is not suitable for publication in Nature Communications.

Specific comments/questions:

- 1) Once again, I want to commend the authors on their careful data analysis and detailed reporting of experimental conditions. The discussion of optical methods in the SI is appreciated. Control experiments (crushing flakes, different atmosphere, optical healing of BE) are particularly confidence-inspiring.
- 2) The key piece of new experimental evidence is the observation of spectrally similar BE even when the sample is excited far below the band gap. As far as I can tell, these data are only reported for the FPEA "red flakes" at room temperature. Is below-gap PL observed in the "green flakes" at low temperature? What about in PEA?
- 3) The authors seem to suggest that the assignment of BE to STE was made casually without careful consideration. I contend that STEs have a long history and that there is indeed strong theoretical support for their occurrence in lead halides. See, for example, <https://doi.org/10.1021/acs.jpcclett.8b03717> (not cited). I think that the last line in the abstract "...hope to stimulate more care in the interpretation of spectroscopic features in this class of materials" is particularly inappropriate.
- 4) The authors should specify which materials they studied in the abstract and explicitly state what new experimental evidence they report.
- 5) I particularly liked the discussion on pages 4-5 introducing STEs and comparing/contrasting to other semiconductors. Again, I think the authors have a number of valid points worth sharing with the community, but the new experimental evidence provided here is insufficient to substantiate a high-profile rebuttal of the STE hypothesis.

Reviewer #2 (Remarks to the Author):

In this work, Kahmann and coworkers are trying to uncover the origin of low-energy broadband emissions observed in 2D perovskite single crystals with both narrow and broadband emissions using various spectroscopic measurements. They prove that the intrinsic self-trapping and surface states are not responsible for the broad emissions of 2D perovskite single crystals (PEA₂PbI₄ and FPEA₂PbI₄), and attribute their broadband emissions to the defect states. They also propose the possible mechanism for the broadband emissions of 2D hybrid perovskites, which is useful for designing the low-dimensional perovskites with efficient broadband emissions. The manuscript is well organized and

written. While I would suggest the authors change the title “Self-trapped excitons in low-dimensional perovskites - do they exist?”. Of course, the self-trapped excitons exist in the most low-dimensional perovskites with only broadband emission that the crystal lattices are easily distorted by photoexcitation. However, in this work, the authors only exclude the self-trapped excitons in two specific 2D perovskite single crystals that have dual emissions. Therefore, I may recommend its publication in Nature Communications after addressing this and below major issues:

1. The stacking/aggregation of organic cations, i.e., the strong pi-pi interactions between PEA or between FPEA, may lead to such broadband emissions. Actually, this can be supported by the increased broadband PL intensity after extra PEAI treatment.
2. The authors mention that iodine-related defects plays a decisive role in the 2D compounds. If it is the case, the PEAI treatment can largely reduce such defects, especially iodine vacancy, and enhance the green emission due to the suppression of defect state generation. By contrast, they observed the increased broadband intensity after adding PEAI. Moreover, I would suggest authors clarify the point defects that really contribute to the broadband emissions. The theoretical calculations may help understand the nature of defect states, like defect formation energies and charge transient levels in those 2D perovskites.
3. The authors already clarify the origin of broadband emission in 2D hybrid perovskites with dual emissions. How about the 2D hybrid perovskites only showing broadband emission, such as (EDBE)PbBr₄? Does the defect states also contribute to their broadband emission?
4. What is the PL quantum yield (PLQY) of green-emissive crystal flakes (1) with the strong red emission band after PEI treatment and (2) with the suppressed red emission band? The comparison of PLQY values between these two samples may provide further evidence about the origin of broadband emissions.

Reviewer #3 (Remarks to the Author):

The manuscript Self-trapped excitons in low-dimensional perovskites – do they exist? by Loi et al. is an interesting study into the photophysics of 2D lead iodide halide perovskites that attempts to explain the broad Stokes-shifted photoluminescence as the result of defect states rather than self-trapped excitons. Unfortunately, without significant additional characterization of the materials studied in the manuscript, I do not believe the authors’ interpretation of their results is supported by the data.

Other than photoluminescence measurements, there is no structural or optical characterization of the PEA or FPEA materials studied. PEA has been widely studied with numerous published crystal structures available at multiple temperatures. However, I am only aware of a single study on FPEA, which is cited in the manuscript. This previously published study has no structural characterization available. Given prior work (Smith, M. D.; Jaffe, A.; Dohner, E. R.; Lindenberg, A. M.; Karunadasa, H. I. Structural Origins of Broadband Emission from Layered Pb–Br Hybrid Perovskites. *Chem. Sci.* 2017, 8, 4497–4504) that correlates structural parameters with the existence of broad low-energy PL in 2D halide perovskites, characterization by single crystal diffraction must be done on FPEA, though this should not be difficult given that the authors grew single crystals of FPEA. The FPEA cation is much wider than the PEA cation, which will likely cause significant structural distortion, which is evident from the blue-shifted PL of FPEA compared to PEA (Knutson, J. L.; Martin, J. D.; Mitzi, D. B. Tuning the Band Gap in Hybrid Tin Iodide Perovskite Semiconductors Using Structural Templating. *Inorg. Chem.*

2005, 44, 4699–4705). An indexed powder diffraction pattern should also be provided for PEA to confirm that the authors' material is identical to the material previously studied. In addition, optical absorption or diffuse reflectance measurements are necessary to contextualize the PL results. If (Fig 5C) the broad emission is caused by traps that can be directly excited, this should be able to be seen by steady-state absorption, or if the absorption is too weak to be seen in steady-state by transient absorption measurements.

The authors must also give the thickness of every flake measured optically. 2D Pb-I halide perovskites have an extremely small Stokes shift between excitonic absorption and PL, so self-absorption effects in PL will be very significant and may greatly skew the results. In Figures 2 and 3, are the "green" and "red" flakes identical in thickness? A thicker flake will have much greater self-absorbance of the narrow excitonic PL resonance while the broad emission will not see any self-absorbance, and this difference can cause the difference in intensities shown in Figure 3C. Without a thickness map corresponding to the images shown in Figures 2B-C, 3A-B, and several SI images, I do not believe the results are because of differences in defect density. The AFM image shown in Figure S1 is not enough because it is only on a single flake in isolation. Figure S5 shows "crushed" crystals, but I am not sure what this means. The particle size is still quite large and may also be dominated by the layer thickness. If the crystals are ground to even smaller particle size and still show these distinctions, I would take that as evidence that the "green" and "red" crystals are actually different materials. Do unexfoliated crystals still show the red/green color distinction?

I am also concerned by the data in Figure 4. The authors assume that the "red" and "green" flakes of FPEA are identical in composition. However, the energy of the excitonic PL resonance is different at room temperature, being approximately 2.35 eV for the red flake and 2.4 eV for the green flake. However, in Figure 3C, the maximum of the excitonic PL resonance is identical for the green and red flakes. This could be caused by self-absorption effects, but it's impossible to know without knowing the thickness of the flakes measured. Furthermore, since these are single crystals and the flakes are on the order of hundreds of microns, they are of sufficient size for single-crystal X-ray diffraction measurements, which will conclusively demonstrate whether the "green" and "red" flakes have the same crystal structure.

I have several other minor concerns. First, the cryogenic measurements are conducted in an evacuated cold-finger cryostat on quartz substrates. Quartz is a thermal insulator and is not suitable as a substrate for cryogenic measurements, so I am concerned that the reported temperature is much higher than the sample temperature. I do not see splitting of the main excitonic PL resonance into several features at temperatures <75K, as has been previously reported in PEA (for instance, Neutzner, S.; Thouin, F.; Cortecchia, D.; Petrozza, A.; Silva, C.; Srimath Kandada, A. R. Exciton-Polaron Spectral Structures in Two-Dimensional Hybrid Lead-Halide Perovskites. *Phys. Rev. Mater.* 2018, 2, 064605 and Straus, D. B.; Iotov, N.; Gau, M. R.; Zhao, Q.; Carroll, P. J.; Kagan, C. R. Longer Cations Increase Energetic Disorder in Excitonic 2D Hybrid Perovskites. *J. Phys. Chem. Lett.* 2019, 10, 1198–1205) and I would expect to occur in FPEA. Sapphire or another highly thermally conductive substrate must be used instead.

I also find figure 5C is misleading. What is the power density in photons/cm² and (once the thickness of the flakes is known) in photons/cm³? It's possible that when focusing to a diffraction-limited spot size, that the instantaneous photon flux is still sufficiently high for nonlinear processes even under CW and picosecond pulsed excitation. I am also not convinced by this data because of the difference in wavelengths: the ultrafast excitation is at 1.55 eV in which there is no BE PL, but the picosecond is at 1.8 eV where there is significant BE PL. Running the ultrafast Coherent Mira in non-mode-locked CW mode would be one way to get CW excitation at the same wavelength as the ultrafast excitation and would be more convincing. Also, the PL spectra under the excitation conditions

in Figure 5C need to be shown, not just the power dependence of the integrated counts.

A very minor issue is that the characterization that "PEA typically only exhibits the NE in thin films.[25,44]" This is true only at room temperature, not at cryogenic temperatures. As far as I am aware, the broad emission appears in every published spectrum taken at temperatures below ~50K as long as the films are deposited on thermally conducting substrates (not glass or quartz), and I am not aware of the broad emission being present at room temperature in any published measurements.

Overall, I think this is an innovative study that if appropriately revised would be an excellent contribution to Nature Communications.

Reviewer #1: Loi & co-workers report an interesting and carefully-executed study of broad emission (BE) in two A₂PbI₄ perovskites: PEA₂PbI₄ and a mono-fluorinated variant thereof (“FPEA”). Through a wide array of photoluminescence characterization techniques, they conclude that BE in FPEA originates from a distribution of static defect states – rather than a self-trapped exciton (STE).

I agree with the authors’ assignment of broad emission in their FPEA samples to defect states and commend them on their careful spectroscopic characterization. However, the findings reported in this manuscript are not nearly generalizable enough to call into question the STE assignment of temperature-dependent BE in other 2D lead halide perovskites. (I want the authors to know that I write this as someone intimately familiar with the spectroscopic characterization of these materials but who has not previously published on STEs; I have no stake in the assignment of the broad spectral feature either way). Once the claims of the paper are scaled back, it could be published in a more specialized journal. However, it is not suitable for publication in Nature Communications.

Our response: We are grateful for the Reviewer to take his/her time to assess our manuscript and to give such careful feedback. We hope that by addressing his/her comments below and by implementing the suggestions made by the other two reviewers, we have improved our manuscript in such a way to achieve his/her approval for publication.

Reviewer #1: Specific comments/questions:

1) Once again, I want to commend the authors on their careful data analysis and detailed reporting of experimental conditions. The discussion of optical methods in the SI is appreciated. Control experiments (crushing flakes, different atmosphere, optical healing of BE) are particularly confidence-inspiring.

Our response: We are especially grateful for the Reviewer to acknowledge and highlight our efforts to disclose all relevant experimental conditions that will allow further groups to check our results or apply our techniques on different systems. We furthermore appreciate his/her mentioning of the importance of our control experiments in the SI, which we consider to be an integral part of our systematic approach to exclude alternative explanations.

Reviewer #1: 2) The key piece of new experimental evidence is the observation of spectrally similar BE even when the sample is excited far below the band gap. As far as I can tell, these data are only reported for the FPEA “red flakes” at room temperature. Is below-gap PL observed in the “green flakes” at low temperature? What about in PEA?

Our response: The general behaviour of the PEA flakes is identical to the FPEA samples. For clarity, we still focus the bulk of the data in the main text on the latter compound, but added data on below gap excitation of PEA in Figure 5 (c) and S18 of the SI.

As the terms “green” and “red flake” are concepts used by us to describe areas of pronounced green or pronounced red emission, their difference is merely qualitative. Observing the BE upon sub-gap excitation is thus a matter of measurement sensitivity. The data in Fig. S18, for example, shows an intermediate case between “green” and “red” flake, for which we can still clearly excite the BE with low energy light. Since “green flakes” offer increasingly strong BE at low temperature (“become red”), sub-gap excitation will also become observable.

Reviewer #1: 3) The authors seem to suggest that the assignment of BE to STE was made casually without careful consideration. I contend that STEs have a long history and that there is indeed strong theoretical support for their occurrence in lead halides. See, for example, <https://doi.org/10.1021/acs.jpcclett.8b03717> (not cited). I think that the last line in the abstract “...hope to stimulate more care in the interpretation of spectroscopic features in this class of materials” is particularly inappropriate.

Our response: We entirely agree with the Reviewer that the concept of STEs has a long history and was especially used in the context of lead halides – both aspects are stated on page 3 with references [4-21]. We also do not intend to downplay theoretical evidence, the presence of which we acknowledge. To avoid any confusion, we reworked the paragraph on page 3 starting with “*There is indeed [...]*” and also included the reference (new ref [24]) kindly provided by the Reviewer. Also the abstract was reworked.

Reviewer #1: 4) The authors should specify which materials they studied in the abstract and explicitly state what new experimental evidence they report.

Our response: We rewrote the abstract accordingly.

Reviewer #1: 5) I particularly liked the discussion on pages 4-5 introducing STEs and comparing/contrasting to other semiconductors. Again, I think the authors have a number of valid points worth sharing with the community, but the new experimental evidence provided here is insufficient to substantiate a high-profile rebuttal of the STE hypothesis.

Our response: We thank the Reviewer for the positive comments on our introduction section, as we indeed deemed it particularly important to discuss how the STE concept arose and how the two phenomena can be discriminated. Again, our point is not to state that STE do not exist, but that care should be taken in the assignment and that not every broad emission feature is an STE.

Reviewer #2: In this work, Kahmann and co-workers are trying to uncover the origin of low-energy broadband emissions observed in 2D perovskite single crystals with both narrow and broadband emissions using various spectroscopic measurements. They prove that the intrinsic self-trapping and surface states are not responsible for the broad emissions of 2D perovskite single crystals (PEA₂PbI₄ and FPEA₂PbI₄), and attribute their broadband emissions to the defect states. They also propose the possible mechanism for the broadband emissions of 2D hybrid perovskites, which is useful for designing the low-dimensional perovskites with efficient broadband emissions. The manuscript is well organized and written.

Our response: We thank the Reviewer for taking the time to study our manuscript, the positive assessment and for his/her detailed comments and questions.

Reviewer #2: While I would suggest the authors change the title “Self-trapped excitons in low-dimensional perovskites - do they exist?”. Of course, the self-trapped excitons exist in the most low-dimensional perovskites with only broadband emission that the crystal lattices are easily distorted by

photoexcitation. However, in this work, the authors only exclude the self-trapped excitons in two specific 2D perovskite single crystals that have dual emissions. Therefore, I may recommend its publication in Nature Communications after addressing this and below major issues:

Our response: We thank the reviewer for the suggestion. As stated above, we changed the title. The Reviewer here refers to the issue of the generalisation of our observations. He/She is certainly correct that we only studied a small selection of materials, which are of a two-dimensional nature. Since we disproved the common theory for these materials, we expect this will spark controversies and possible re-evaluations of previous results including previous attribution to luminescence of STEs – in particular, for such 2D compounds that seemingly only offer the BE and for compounds of lower dimensionality.

Reviewer #2: 1. The stacking/aggregation of organic cations, i.e., the strong pi-pi interactions between PEA or between FPEA, may lead to such broadband emissions. Actually, this can be supported by the increased broadband PL intensity after extra PEAI treatment.

Our response: Unfortunately, this comment is unclear to us. The Reviewer seems to suggest that the aggregation of the molecules led to the broad emission as if they were the excited and emitting species. Since the HOMO-LUMO gap of the molecules is much wider than the employed 3.1 eV photon energy, this is not possible.

Reviewer #2: 2. The authors mention that iodine-related defects play a decisive role in the 2D compounds. If it is the case, the PEAI treatment can largely reduce such defects, especially iodine vacancy, and enhance the green emission due to the suppression of defect state generation. By contrast, they observed the increased broadband intensity after adding PEAI. Moreover, I would suggest authors clarify the point defects that really contribute to the broadband emissions. The theoretical calculations may help understand the nature of defect states, like defect formation energies and charge transient levels in those 2D perovskites.

Our response: We thank the Reviewer for these suggestions. Theoretical work on defects in 2D perovskites has been carried out before, e.g. in 10.1021/acs.nanolett.6b00964, which we added as the new reference [53]. The authors of this work describe how different environments during synthesis can easily create interstitial halides or A-site ions that create states deep in the band gap. The corresponding passage on page 17 was updated accordingly.

Furthermore, we have become aware of a separate study involving the synthesis of PEA₂PbI₄ under different concentrations of iodide precursor and its impact on the BE (10.1021/acs.jpcclett.9b00934, new ref[49]), in which the authors also propose a connection between iodide interstitials and the BE.

The updated section on page 17 now reads: *“Especially for the 3D counterparts of these materials, the (light-mediated) diffusion of ions, particularly methylammonium and halide ions, has been associated with changes in luminescence spectra and dynamics.[51,52] Given the absence of the former, we consider it probable, that iodide-related (more generally, halide-related) defects, such as vacancies or interstitials play a decisive role also in the 2D compounds. Especially the latter are proposed to form deep states[53] and were previously linked to the broad emission.[49]”*

Reviewer #2: 3. The authors already clarify the origin of broadband emission in 2D hybrid perovskites with dual emissions. How about the 2D hybrid perovskites only showing broadband emission, such as (EDBE)PbBr₄? Does the defect states also contribute to their broadband emission?

Our response: Currently, we think the emergence of solely a BE should be a consequence of very efficient transitions towards the trap states (similar to our bright red flakes), so that there is merely a qualitative difference in the systems, but to probe this behaviour we will have to identify several appropriate candidate materials for synthesis. Since our main goal in the manuscript is to spark a discussion on the relevance of STEs and to raise awareness of alternative explanations, we do not aim to investigate all relevant systems ourselves.

Reviewer #2: 4. What is the PL quantum yield (PLQY) of green-emissive crystal flakes (1) with the strong red emission band after PEI treatment and (2) with the suppressed red emission band? The comparison of PLQY values between these two samples may provide further evidence about the origin of broadband emissions.

Our response: Here we need to disagree with the Reviewer. PLQY measurements are indeed very interesting in their own right, especially when aiming e.g., for applying these materials in lighting applications, but we are of the opinion that they do not serve to elucidate the origin of the BE.

Reviewer #3: The manuscript Self-trapped excitons in low-dimensional perovskites – do they exist? by Loi et al. is an interesting study into the photophysics of 2D lead iodide halide perovskites that attempts to explain the broad Stokes-shifted photoluminescence as the result of defect states rather than self-trapped excitons. Unfortunately, without significant additional characterization of the materials studied in the manuscript, I do not believe the authors' interpretation of their results is supported by the data.

Other than photoluminescence measurements, there is no structural or optical characterization of the PEA or FPEA materials studied. PEA has been widely studied with numerous published crystal structures available at multiple temperatures. However, I am only aware of a single study on FPEA, which is cited in the manuscript. This previously published study has no structural characterization available. Given prior work (Smith, M. D.; Jaffe, A.; Dohner, E. R.; Lindenberg, A. M.; Karunadasa, H. I. Structural Origins of Broadband Emission from Layered Pb–Br Hybrid Perovskites. Chem. Sci. 2017, 8, 4497–4504) that correlates structural parameters with the existence of broad low-energy PL in 2D halide perovskites, characterization by single crystal diffraction must be done on FPEA, though this should not be difficult given that the authors grew single crystals of FPEA. The FPEA cation is much wider than the PEA cation, which will likely cause significant structural distortion, which is evident from the blue-shifted PL of FPEA compared to PEA (Knutson, J. L.; Martin, J. D.; Mitzi, D. B. Tuning the Band Gap in Hybrid Tin Iodide Perovskite Semiconductors Using Structural Templating. Inorg. Chem. 2005, 44, 4699–4705). An indexed powder diffraction pattern should also be provided for PEA to confirm that the authors' material is identical to the material previously studied.

Our response: We thank the Reviewer for his/her assessment of our manuscript and the detailed comments and concerns raised. Here the Reviewer correctly points out the broad literature on PEA-based materials and the virtual non-existence of reports on FPEA and suggests that structural origins might be responsible for the BE in FPEA.

We would like to underline again that although the bulk of our manuscript shows data on FPEA, the findings on the BE are identical for both compounds, as we stress along the main text and show in

the Figures. 2, 5, S2, S4, S8, S17, S19). Structural differences between the two compounds are thus irrelevant for the emergence of the BE.

Nonetheless, we would like to disclose that we currently prepare a separate in-depth study on the properties of FPEA₂PbI₄ comparing it to PEA₂PbI₄. In our view, an in-depth structural characterisation of FPEA does not help with the current task of discriminating between STEs and traps. For the Reviewer's consideration, we can nonetheless disclose that based on x-ray diffraction measurements, for which a pattern of the $hk0$ plane is shown below, the crystal structure of FPEA could be solved in the $I2/m$ space group. As the confocal micrographs next to it indicate, these measurements were carried out on flakes that exhibited both NE and BE emission and we did not observe any indication of phase impurity. Similarly, we also determined the crystal structure of the PEA flakes, which is in line with reports in literature.

Furthermore, the Reviewer correctly points out that FPEA shows a blue shifted NE compared to PEA and that organic spacers can have a templating effect on the distortion of the octahedral cages. We found that for FPEA the average Pb-I-Pb bond angle is 150.25 degrees showing a larger distortion than the Pb-I-Pb bond angle of 154.05 degrees for PEA. Again, these are interesting aspects, which we shall publish, but as we do not see a clear connection to the discussion of this manuscript, we intend to report the data in a separate manuscript.

Finally, as mentioned above, we highlight in our reworked manuscript that the BE can be tuned by the precursor ratio of cast thin films, which is additional evidence that crystal structure is not the important property to explain the BE, but that it is defect related. We recently discussed the importance of the precursor stoichiometry on the photoluminescence properties in 10.1002/adfm.201907505 (ref [42]), which contains the following graph as Figure S5:

Here, the PL intensity of different PEA₂PbI₄ thin films was measured and the precursor variation clearly affects the presence/absence of the BE, while no difference, e.g. in the crystal structure or NE position could be observed. We added similar data to Figure S20. Moreover, the newly added ref [49] reports similar findings for PEA₂PbI₄ upon variation of the iodide precursor concentration.

Reviewer #3: In addition, optical absorption or diffuse reflectance measurements are necessary to contextualize the PL results. If (Fig 5C) the broad emission is caused by traps that can be directly excited, this should be able to be seen by steady-state absorption, or if the absorption is too weak to be seen in steady-state by transient absorption measurements.

Our response: For crystal flakes the below gap absorption signal is generally governed by significant scattering and even for our thin film samples direct optical absorption is not applicable – the Reviewer rightly suggests that the signals are too weak. Photoinduced absorption (PIA) spectroscopy on similar materials has indeed been carried out before (e.g. in 10.1021/ja512833n, ref [29]) and photoinduced absorption bands at below gap energy were found. Nonetheless, since PIA as a technique is unable to discriminate whether such bands are due to trap states or STEs, we deem it not helpful for our discussion.

Reviewer #3: The authors must also give the thickness of every flake measured optically. 2D Pb-I halide perovskites have an extremely small Stokes shift between excitonic absorption and PL, so self-absorption effects in PL will be very significant and may greatly skew the results. In Figures 2 and 3, are the “green” and “red” flakes identical in thickness? A thicker flake will have much greater self-absorbance of the narrow excitonic PL resonance while the broad emission will not see any self-absorbance, and this difference can cause the difference in intensities shown in Figure 3C. Without a thickness map corresponding to the images shown in Figures 2B-C, 3A-B, and several SI images, I do not believe the results are because of differences in defect density. The AFM image shown in Figure S1 is not enough because it is only on a single flake in isolation. Figure S5 shows “crushed” crystals, but I am not sure what this means. The particle size is still quite large and may also be dominated by the layer thickness. If the crystals are ground to even smaller particle size and still show these distinctions, I would take that as evidence that the “green” and “red” crystals are actually different materials. Do unexfoliated crystals still show the red/green color distinction?

I am also concerned by the data in Figure 4. The authors assume that the “red” and “green” flakes of FPEA are identical in composition. However, the energy of the excitonic PL resonance is different at room temperature, being approximately 2.35 eV for the red flake and 2.4 eV for the green flake. However, in Figure 3C, the maximum of the excitonic PL resonance is identical for the green and red flakes. This could be caused by self-absorption effects, but it’s impossible to know without knowing the thickness of the flakes measured. Furthermore, since these are single crystals and the flakes are on the order of hundreds of microns, they are of sufficient size for single-crystal X-ray diffraction measurements, which will conclusively demonstrate whether the “green” and “red” flakes have the same crystal structure.

Our response: Yes, unexfoliated crystals show the same distinction of red and green colour. We chose to focus on exfoliated crystals to exclude surface effects through sample deterioration.

The Reviewer raises the very important issue of self-absorption and proposes that the intensity differences might be due to thickness effects. First we would like to note that, as stated in the experimental section, we measured the PL in reflection mode, which suppresses possible thickness effects to a minimum. Nonetheless, we entirely agree that the small Stokes shift of the NE can have a big impact on the observed peak position. The two-photon spectra in Fig. S19, for example, show a red-shifted peak for PEA_2PbI_4 . Since in the updated temperature dependent spectra of Figure 4 the NE peaks at the same energy for both the red or green flake and we generally did not observe such a shift in other measurements, we assume that the previous discrepancy was more likely a measurement artefact.

Nonetheless, we disagree on the point of thickness being the origin of the NE/BE intensity variation. This is chiefly for three reasons as elucidated in the following.

Firstly, the Reviewer is correct, of course, to state that the BE will not undergo significant reabsorption and will thus appear more pronounced compared to the NE when both signals travel the same distance within the crystal. If this were the sole reason for the intensity difference of our samples, this would also mean that all crystal edges would appear bright red in wide field PL microscopy (the lateral extend of the flakes is orders of magnitude larger than their thickness). As shown for example in Fig. 2 (c) or Fig. S3, this is not the case.

Secondly, assuming reabsorption to be the origin of the NE/BE variation is incompatible with the transient PL data we discuss in Fig. S12. Red flakes exhibit a distinctly shorter PL lifetime of the NE, which is evidence for the presence of additional non-radiative decay channels for the NE where the BE is observed. Thickness effects cannot explain this observation.

Thirdly, as stated above, the BE can be suppressed or amplified in thin films through precursor variation without any dependence on the thickness (ref [42], figure above).

We can thus conclusively exclude the flake thickness as dominant process for the observation of the BE and state as much on page 11 (top).

Reviewer #3: I have several other minor concerns. First, the cryogenic measurements are conducted in an evacuated cold-finger cryostat on quartz substrates. Quartz is a thermal insulator and is not suitable as a substrate for cryogenic measurements, so I am concerned that the reported temperature is much higher than the sample temperature. I do not see splitting of the main excitonic PL resonance into several features at temperatures $<75\text{K}$, as has been previously reported in PEA (for

instance, Neutzner, S.; Thouin, F.; Cortecchia, D.; Petrozza, A.; Silva, C.; Srimath Kandada, A. R. Exciton-Polaron Spectral Structures in Two-Dimensional Hybrid Lead-Halide Perovskites. *Phys. Rev. Mater.* 2018, 2, 064605 and Straus, D. B.; Iotov, N.; Gau, M. R.; Zhao, Q.; Carroll, P. J.; Kagan, C. R. Longer Cations Increase Energetic Disorder in Excitonic 2D Hybrid Perovskites. *J. Phys. Chem. Lett.* 2019, 10, 1198–1205) and I would expect to occur in FPEA. Sapphire or another highly thermally conductive substrate must be used instead.

Our response: The Reviewer raises an important issue here. Since in our geometry the entire substrate is connected to a copper plate, the thermal contact is generally very good and we observe the above mentioned splitting for thin films of PEA2PbI4 using the exact same measurement conditions. However, as the flakes were pressed onto the quartz substrate, we assumed there could have been insufficient heat conduction through the quartz/flake interface. We therefore carried out the key measurements again using a different cryostat working with the sample stored in helium atmosphere (the experimental section was updated accordingly).

The results show that the contact was indeed insufficient and the mentioned splitting of the peaks can now be observed (especially pronounced for PEA). We consequently changed all data in Figure 4 of the main text as well as S13-17 of the Supporting Information. The main trend of the brightening BE at low temperature as well as the different impact of cooling for the red and green flake remain unaffected. A new finding, however, is that below approximately 80 K, all samples show the formation of the intermediate state emission around 2.15 eV. The section in the main text and SI was updated accordingly.

Reviewer #3: I also find figure 5C is misleading. What is the power density in photons/cm² and (once the thickness of the flakes is known) in photons/cm³? It's possible that when focusing to a diffraction-limited spot size, that the instantaneous photon flux is still sufficiently high for nonlinear processes even under CW and picosecond pulsed excitation. I am also not convinced by this data because of the difference in wavelengths: the ultrafast excitation is at 1.55 eV in which there is no BE PL, but the picosecond is at 1.8 eV where there is significant BE PL. Running the ultrafast Coherent Mira in non-mode-locked CW mode would be one way to get CW excitation at the same wavelength as the ultrafast excitation and would be more convincing. Also, the PL spectra under the excitation conditions in Figure 5C need to be shown, not just the power dependence of the integrated counts.

Our response: Following the Reviewer's suggestion, we converted the x-axis into a photon fluence in photons/cm² s. We also noticed that the original Figure contained a typographic error – the below gap excitation was falsely indicated as 690 nm, but was actually at 600 nm. For consistency reasons we furthermore converted all wavelengths into photon energies in eV.

However, we think there might have been a misunderstanding on the Reviewer's side of what Figure 5 (c) is supposed to convey. The crucial aspect of Fig. 5(c) is to exclude two-photon absorption for the spectra in 5 (b). This was done by showing the linear behaviour of the BE intensity. The Reviewer states that at 1.55 eV (800 nm) there was no BE PL visible, but this is incorrect. The red triangles indicate the BE upon 1.55 eV excitation, as also discussed on page 14 (centre) in the main text. The important aspect is that it scales with an exponent of 2 (just like the NE), which indicates that PL in this case is due to two-photon absorption. The only reason we show the 1.55 eV data here is to exemplify how 1- and 2-photon excitation can be discriminated – the 1.55 eV data are irrelevant to our argument. In this light it is unclear to us what the proposed cw-excitation at 1.55 eV ought to reveal.

In order to clarify our message, we reworked the caption of Figure 5 and the corresponding part of the main body of the text on page 14. We furthermore now include BE data on both flakes in (c) and, following the Reviewer's suggestion, we also added full spectra and an example highlighting the necessity of the employed filters to the Supporting Information in Fig. S18. Since this is the key experiment in our manuscript, we also added further clarifying remarks to the SI on page 21 as well as additional data for the two-photon process in Fig. S19.

We hope that these changes as well as our explanations above convince the Reviewer of the soundness of our data.

Reviewer #3: A very minor issue is that the characterization that "PEA typically only exhibits the NE in thin films.[25,44]" This is true only at room temperature, not at cryogenic temperatures. As far as I am aware, the broad emission appears in every published spectrum taken at temperatures below ~50K as long as the films are deposited on thermally conducting substrates (not glass or quartz), and I am not aware of the broad emission being present at room temperature in any published measurements.

Our response: We agree with the Reviewer that PEA tends to show the BE at low temperature. At this point in the manuscript we were implicitly only referring to behaviour at room temperature, but did not state so explicitly. This was changed and the statement in question now reads: "*PEA typically only exhibits the NE at room temperature.*"

Nonetheless, although not common, the BE has indeed been observed for PEA at room temperature before. For example reference [27] includes it (10.1021/jacs.6b08175), but chiefly discusses this aspect in the SI. Furthermore, our measurements in [42] (10.1002/adfm.201907505) discussing the impact of the precursor stoichiometry were made at room temperature.

Reviewer #3: Overall, I think this is an innovative study that if appropriately revised would be an excellent contribution to Nature Communications.

Our response: We thank the Reviewer dearly and hope to have addressed his/her concerns appropriately.

Reviewers' comments:

Reviewer #1 (Remarks to the Author):

I am appreciative of the time the authors have spent revising their work. However, I still do not support publication in Nature Communications.

Overall, this is a nice study that will stimulate thought but it's not nearly comprehensive enough to justify the title or tone of the abstract. In particular, the breadth of materials systems studied is not broad enough to warrant such sweeping claims.

The debate over STE vs defects is not new. Writing in Chemical Reviews last year (<http://dx.doi.org/10.1021/acs.chemrev.8b00477>, not cited), Karunadasa & co-workers attempted to reconcile divergent observations in the literature surrounding the choice of halide anion: "7.3.2. An Attempt at Unifying the Mechanism of the Broad Emission from Pb–X Perovskites (X = Cl, Br, and I). Mechanistic studies show indications of transient, light induced defects in Pb–Br perovskites that emit white light,⁹⁰ and permanent light-induced defects in Pb–I perovskites that exhibit a broad emission.²⁰⁹ ... We can thus attempt to unify the separate observations for Pb–Cl, Pb–Br, and Pb–I perovskites by considering that the transient defects created through self-trapping can evolve into permanent material defects, as previously suggested.⁸⁸ This process is likely facile in iodide perovskites, and may require greater excitation fluences in bromide and chloride perovskites. This phenomenon is not new; self-trapping precedes the formation of permanent lattice defects in related materials, such as the formation of color centers (electron trapped in a halide vacancy) in alkali metal halides.¹⁵⁸"

Karunadasa's main point is that defects and self-trapped excitons are closely related phenomena, and that iodides (as studied by Loi & co-workers here) are the most likely to exhibit emission from permanent traps. I regret that I didn't raise this point in my first round of review, but I was not aware of this text at that time.

I see the present manuscript by Loi & co-workers as a valuable data point among many; not a breakthrough. I believe the authors' interpretation of their data and I think the below-gap excitation experiments are particularly novel, but the scope of work is limited to only two n=1 iodide perovskites. It's simply not strong enough evidence to call into question a whole body of literature. I would support publication of this work in a more specialized journal like J Phys Chem Lett, but the broad conclusions are not sufficiently supported by the limited data set to warrant publication in Nature Communications.

Reviewer #2 (Remarks to the Author):

In the revised manuscript, the authors have addressed my concerns by changing the title and providing more detailed discussions of possible origin of defect states. Based on these modifications, the conclusions are now convincing, showing the broadband emission of studied 2D hybrid perovskite crystals originates from the defect states instead of STEs. Therefore, I recommend the current version could be published in Nature Communications without further review.

Reviewer #3 (Remarks to the Author):

I thank the authors for the detailed responses to my comments and for the change they have made.

There are a couple issues that remain before I can recommend publication, some of which are in the new data included in this revision.

First, I cannot recommend publication unless the crystal structure CIF file for FPEA and details of its collection and refinement are included with the manuscript. No structural details of this compound are reported in the literature, and while I respect the desire to have a separate publication on this compound, it is impossible to contextualize the data in this manuscript without the full structural parameters of this compound published with this manuscript. To elaborate, inspection of the $hk0$ plane shown in the authors' response to my initial review shows evidence of significant twinning based on the splitting of spots, but I cannot say for certain without the structure and details of the refinement. The splitting as well as other stray reflections may be significant and indicate the presence another phase or something unexpected, or they may instead be artifacts created in the generation of the precession image shown.

Second, no characterization of PEA is included, and the authors need to at least include a powder X-ray diffraction pattern of their PEA sample. This needs to be included to confirm there are no significant impurities in the authors' PEA compared to what has been measured previously, given the emphasis placed on the intensity of the different PL resonances. Is it possible any of these PL resonances are caused by discrete impurities (for instance, see below about the high energy PL in PEA)? The paper would be more convincing with this additional structural analysis.

Third, what is the origin of the peak at 2.5 eV in FPEA below ~ 10 K (Figure 4C-D)? What does the thin film absorption spectrum look like? Is there a phase transition? Reports of excitonic structure in cryogenic absorption and PL spectra of PEA_2PbI_4 thin films show resonances spaced by 30-40 meV, but here this looks to be ~ 100 -200 meV blue of the main NE peak, which is too large to be vibronic. Is this a phase transition? Also, the hot PL in PEA_2PbI_4 is different here than previously reported with features much higher in energy at different energetic spacing (e.g., Neutzner, S.; Thouin, F.; Cortecchia, D.; Petrozza, A.; Silva, C.; Srimath Kandada, A. R. Exciton-Polaron Spectral Structures in Two-Dimensional Hybrid Lead-Halide Perovskites. *Phys. Rev. Mater.* 2018, 2, 064605 and D. B. Straus, S. Hurtado Parra, N. Iotov, J. Gebhardt, A. M. Rappe, J. E. Subotnik, J. M. Kikkawa, C. R. Kagan, J. Am. Chem. Soc. 2016, 138, 13798.). Some comment should be made on why the PL spectrum appears different here. Perhaps it is because these are crystals compared to the films measured in those papers, but it is not obvious to me why that should matter for PL higher in energy than the main excitation peak.

I still think this is a good fit for Nature Communications, but the above comments still need to be addressed before I can recommend publication.

Reviewer #1: I am appreciative of the time the authors have spent revising their work. However, I still do not support publication in Nature Communications.

Our response: We thank the Reviewer for taking the time to re-assess our manuscript and providing constructive feedback.

Overall, this is a nice study that will stimulate thought but it's not nearly comprehensive enough to justify the title or tone of the abstract. In particular, the breadth of materials systems studied is not broad enough to warrant such sweeping claims.

Our response: Given the Reviewer's comment, we re-assessed both the title and the abstract. Whilst we do not see any problem with the title, which is phrased as an open question, not a statement – we made minor changes to the wording of the abstract, where previous phrasing could have been perceived as inappropriate.

Reviewer #1: The debate over STE vs defects is not new. Writing in Chemical Reviews last year (<http://dx.doi.org/10.1021/acs.chemrev.8b00477>, not cited), Karunadasa & co-workers attempted to reconcile divergent observations in the literature surrounding the choice of halide anion: “7.3.2. An Attempt at Unifying the Mechanism of the Broad Emission from Pb–X Perovskites (X = Cl, Br, and I). Mechanistic studies show indications of transient, light induced defects in Pb–Br perovskites that emit white light,⁹⁰ and permanent light-induced defects in Pb–I perovskites that exhibit a broad emission.²⁰⁹ ... We can thus attempt to unify the separate observations for Pb–Cl, Pb–Br, and Pb–I perovskites by considering that the transient defects created through self-trapping can evolve into permanent material defects, as previously suggested.⁸⁸ This process is likely facile in iodide perovskites, and may require greater excitation fluences in bromide and chloride perovskites. This phenomenon is not new; self-trapping precedes the formation of permanent lattice defects in related materials, such as the formation of color centers (electron trapped in a halide vacancy) in alkali metal halides.¹⁵⁸”

Karunadasa's main point is that defects and self-trapped excitons are closely related phenomena, and that iodides (as studied by Loi & co-workers here) are the most likely to exhibit emission from permanent traps. I regret that I didn't raise this point in my first round of review, but I was not aware of this text at that time.

Our response: We agree with the Reviewer that defects and self-trapped excitons can(!) be closely related. We also agree that the given reference discusses the contrast of many bromide and chloride-based layered halide perovskite with a broad emission on the one hand and its relative absence in iodide compounds on the other – an aspect we highlighted in the introduction.

However, we would like to note that this thorough and well-written review of the Karunadasa group does not contain original data on the distinction. The authors merely speculate on the origin of the differences – namely, that illumination creates transient defects (read “STE”) in the bromide/chloride compounds and permanent defects in the latter (as included in the quotation given by the Reviewer).

As we show in our work, the effect of light is the opposite in our materials. Prolonged illumination reduces the intensity of the BE (Fig. S10) and we furthermore highlight the impact of the precursor ratio on the BE through Figure S20, which is independent of any influence of light.

We highlighted this important distinction by inserting the following statement when discussing Figure S10 & S11 on page 11: “[...] (Fig. S10, S11), which excludes it to be due to light-induced defects as previously speculated.[42].”, which includes the new reference [42] to the work of the Karunadasa group.

Whilst we thus agree that experts in the field may have noticed differences in the behaviour of bromide/chloride-based materials vs. iodide-based compounds and speculated their own conclusions on the origin, we here present experimental evidence and thoroughly show what needs to be done to discriminate the two phenomena. In our opinion this is a valuable and needed contribution to the research field.

Reviewer 1: I see the present manuscript by Loi & co-workers as a valuable data point among many; not a breakthrough. I believe the authors’ interpretation of their data and I think the below-gap excitation experiments are particularly novel, but the scope of work is limited to only two $n=1$ iodide perovskites. It’s simply not strong enough evidence to call into question a whole body of literature. I would support publication of this work in a more specialized journal like J Phys Chem Lett, but the broad conclusions are not sufficiently supported by the limited data set to warrant publication in Nature Communications.

Our response: This point is largely addressed above. We thank the Reviewer kindly for underlining that he/she believes our interpretation of the data and that our below band gap experiments are particularly novel and accept that he/she has a different opinion on the relevance of our work. We respectfully disagree on the latter aspect.

Reviewer #2: In the revised manuscript, the authors have addressed my concerns by changing the title and providing more detailed discussions of possible origin of defect states. Based on these modifications, the conclusions are now convincing, showing the broadband emission of studied 2D hybrid perovskite crystals originates from the defect states instead of STEs. Therefore, I recommend the current version could be published in Nature Communications without further review.

Our response: We thank the Review for the positive assessment of our updated manuscript.

Reviewer #3: I thank the authors for the detailed responses to my comments and for the change they have made. There are a couple issues that remain before I can recommend publication, some of which are in the new data included in this revision.

Our response: We thank the Reviewer for taking the time to re-assess our work and address the issues below.

Reviewer #3: First, I cannot recommend publication unless the crystal structure CIF file for FPEA and details of its collection and refinement are included with the manuscript. No structural details of this compound are reported in the literature, and while I respect the desire to have a separate publication on this compound, it is impossible to contextualize the data in this manuscript without the full structural parameters of this compound published with this manuscript. To elaborate, inspection of the hk0 plane shown in the authors' response to my initial review shows evidence of significant twinning based on the splitting of spots, but I cannot say for certain without the structure and details of the refinement. The splitting as well as other stray reflections may be significant and indicate the presence another phase or something unexpected, or they may instead be artifacts created in the generation of the precession image shown.

Second, no characterization of PEA is included, and the authors need to at least include a powder X-ray diffraction pattern of their PEA sample. This needs to be included to confirm there are no significant impurities in the authors' PEA compared to what has been measured previously, given the emphasis placed on the intensity of the different PL resonances. Is it possible any of these PL resonances are caused by discrete impurities (for instance, see below about the high energy PL in PEA)? The paper would be more convincing with this additional structural analysis.

Our response: We understand and appreciate the Reviewer's desire for seeing additional structural data on the compounds (which is why we shared some aspects in our previous response). We would like to underline that we did not split it off to have data for a separate publication, as the Reviewer seems to allege. As mentioned in our first response, we think that its addition to the current (already long) manuscript will serve no constructive purpose – a detailed discussion on the structure refinement and crystallographic measurements will not help in the discrimination of the origin of the BE. As mentioned in our previous answer, the presence of the BE in both compounds underlines that the BE is not of structural origin. Moreover, powder XRD is in our eyes not a viable technique for detecting impurities.

We studied the PEA crystal structure and found it to be identical as what was found *e.g.* by the Mitzi group (<https://doi.org/10.1021/acs.inorgchem.7b01094>).

We would like to ask the Reviewer to consider that our spectroscopic data (*e.g.* NE peak energy & width or PL lifetime) is generally in perfect agreement with the literature and if there were any additional phases in the crystals (which we do not observe), this would be completely in line with our claim that the BE is not due to intrinsic processes of A_2PbI_4 .

Reviewer #3: Third, what is the origin of the peak at 2.5 eV in FPEA below ~10 K (Figure 4C-D)? What does the thin film absorption spectrum look like? Is there a phase transition? Reports of excitonic structure in cryogenic absorption and PL spectra of PEA_2PbI_4 thin films show resonances spaced by 30-40 meV, but here this looks to be ~100-200 meV blue of the main NE peak, which is too large to be vibronic. Is this a phase transition? Also, the hot PL in PEA_2PbI_4 is different here than previously reported with features much higher in energy at different energetic spacing (*e.g.*, Neutzner, S.; Thouin, F.; Cortecchia, D.; Petrozza, A.; Silva, C.; Srimath Kandada, A. R. Exciton-Polaron Spectral Structures in Two-Dimensional Hybrid Lead-Halide Perovskites. *Phys. Rev. Mater.* 2018, 2, 064605 and D. B. Straus, S. Hurtado Parra, N. Iotov, J. Gebhardt, A. M. Rappe, J. E. Subotnik, J. M. Kikkawa, C.

R. Kagan, J. Am. Chem. Soc. 2016, 138, 13798.). Some comment should be made on why the PL spectrum appears different here. Perhaps it is because these are crystals compared to the films measured in those papers, but it is not obvious to me why that should matter for PL higher in energy than the main excitation peak.

Our response: We agree that there is a difference between the 2.5 eV peaks and the previously discussed “hot PL”. The different appearance of our spectra and the ones in the references can be explained relatively easily through the experimental conditions. We used a much coarser grating to simultaneously track the NE and BE – this forbids us to properly resolve the NE sub-structure with the “hot PL”. At the same time, we present a significantly higher dynamic range, which allows for observing the 2.5 eV signals (the given references either cut-off their plots below 2.5 eV or use a linear scale, where a contribution of the magnitude seen in our data would not be resolved).

As to the origin of these peaks, we can only speculate, but we consider it likely that they are due to minor pollution of PbI₂, which can form *e.g.* under intense laser irradiation at the surface (as shown previously by our group 10.1002/adfm.201800305). PbI₂ exhibits excitonic emission around 2.5 eV, which brightens at low temperature (10.1143/JPSJ.51.3228 & 10.1016/0022-3697(69)90146-2). We included a statement about the role of PbI₂ on the peaks at 2.5 eV on page 12 as stated below and also added it to the list of materials that exhibit broad defect-related luminescence. However, at this point we consider it very important to note that from simultaneous studies on thin films we know there is no correlation between the presence of the 2.5 eV emission and the BE – in other words it is not just PbI₂ pollution that gives rise to the BE.

“As a side note, we would like to mention that both types of materials can exhibit PL peaks around 2.5 eV, on the high energy side of the NE (Fig. 4 (c), (d) and Fig. S13). These peaks are different from the previously reported hot PL peaks [27] and we tentatively attribute them to small amounts of PbI₂, [44,45] which can form through extended illumination.[26]

I still think this is a good fit for Nature Communications, but the above comments still need to be addressed before I can recommend publication.